# Novel Plasmonic Metamaterials Based on Metal Nano-Hemispheres and Metal-Dielectric Composites

**Rei Niguma, Tetsuya Matsuyama, Kenji Wada and Koichi Okamoto ***

Department of Physics and Electronics, Osaka Metropolitan University, 1-1 Gakuen-cho,
Naka-ku, Sakai-shi 599-8531, Japan
* Correspondence: okamot@omu.ac.jp; Tel.: +81-72-254-9263

**Abstract:** We introduce a groundbreaking plasmonic metamaterial, the Nano-Hemisphere on Hyperbolic Metamaterial (NHoHMM), which involves the fabrication of Ag nano-hemispheres on a multilayered Ag/SiO$_2$ structure, achieved solely through sputtering and heat treatment. Finite Difference Time Domain (FDTD) simulations unveil the intriguing slow propagation of the localized electric field, where light travels at only 1/40th of its usual speed within this structure. These simulations reveal distinctive sharp absorption peaks in the visible spectrum, attributed to surface plasmon resonance. In practical experiments, the NHoHMM structure, characterized by random Ag nano-hemispheres, exhibits broad absorption peaks spanning the visible range, rendering it a versatile broadband optical absorber. For comparison, the optical properties of the Ag nano-hemispheres on a nanocermet (NHoNC) structure were analyzed through simultaneous sputtering of Ag and SiO$_2$ followed by heat treatment. Simulations employing effective medium theory and the transfer matrix method demonstrate variable optical properties dependent on the Ag filling ratio in the nanocermet structure. The results obtained differ from the spectra of the NHoHMM structure; thus, it is concluded that in the NHoHMM structure, the calculated multi-peaks are broadened due to the inhomogeneity of the nano-hemispherical structure's size, rather than the metal/dielectric multilayer structure being altered by the heat treatment.

**Keywords:** plasmonics; metamaterials; localized surface plasmon resonance; hyperbolic metamaterials; cermet; nanohemisphere-on-mirror

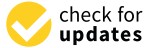



## 1. Introduction

Plasmonic metamaterials utilizing surface plasmons generated at the interface of metals and dielectrics have been an actively researched topic in recent years [1]. These materials have been explored for various applications, including negative refraction [2–9], super-resolution imaging [10–15], and cloaking [16–19], as well as phenomena such as perfect absorption, reflection, slow light [20], and more. However, many of these structures demand extremely high precision in nanofabrication, making them unsuitable for industrial applications due to the substantial cost and time required for large-scale processing. Nevertheless, we have recently reported on the unique optical properties of a three-dimensional multilayer film consisting of Ag nano-sheets arranged on a metal substrate [21,22]. This structure is fabricated using chemical methods and represents one of the plasmonic metamaterials that can be relatively easily produced. Applications such as photocatalytic reaction sensing [23] and enhanced fluorescence imaging [24] have been reported using this structure. Furthermore, we recently introduced a new structure called the Nano-Hemisphere on Mirror (NHoM) [25]. This structure involves placing nano-hemispheres on a metal substrate with dielectric spacer layers, and it can be produced using sputtering and heat treatment alone. With this structure, localized surface plasmon resonance (LSPR) in the deep ultraviolet region [26,27] and high-sensitivity colorimetric sensing [28] have been achieved.

Furthermore, as a metamaterial utilizing surface plasmon resonance, there exists the Hyperbolic Metamaterial (HMM) [29]. An HMM is a nanoscale composite material of metals and dielectrics, often classified into multilayered and three-dimensional structures [30]. In the multilayered structure, sub-wavelength scale metals and dielectrics are periodically stacked, leading to collective vibrations caused by the coupling of surface plasmons excited at individual interfaces. In the three-dimensional structure, metal nanowires are regularly arranged in the dielectric, causing various phenomena by the coupling of surface plasmons generated by these nanowires. The specificity of these HMMs can be represented by the effective medium approximation, treating multilayer films or dielectric/metal nanowire structures as a single effective medium. In this case, the HMM becomes an anisotropic medium with different permittivities in the xy and z directions, and the dispersion relation becomes hyperbolic [30]. By utilizing such properties, various applications beyond the diffraction limit, such as imaging [14,31], photolithography [32,33], biosensors [34], and solar absorption [35], have been proposed. In the realm of composite materials with metals and dielectrics, nanocermet structures are also employed as metamaterials. In nanocermet structures, metal nanoclusters are formed within the dielectric [36], and applications such as broadband absorption have been devised by utilizing this structure [37].

Recent developments in metamaterials have marked significant strides in photonics and electromagnetic control, opening new avenues for applications and improved device functionalities. Hyperbolic metamaterials, for instance, have emerged as pivotal in enhancing optical microscopy and spectroscopy, achieving subwavelength confinement and significant field enhancement to improve resolution in advanced technologies such as "superlens" and surface-enhanced Raman spectroscopy [38]. The exploration of nonlinear optical properties in composite nanomaterials reveals enhanced nonlinearity, attributable to the optical Kerr effect and nonlinear optical absorption, facilitated by advanced pulse techniques [39]. Further, the introduction of an all-dielectric grating absorber with a hemispheric design marks a significant achievement in near-total light absorption and quality factor optimization, validated through analytical modeling and experimental outcomes [40]. The development of ITO/SiO$_2$-based hyperbolic metamaterials introduces electrically tunable dielectric singularities, enhancing light confinement and emission, with promising implications for ultra-fast signal processing and photonic circuit integration [41]. Similarly, a tunable graphene-based hyperbolic microcavity for mid-infrared nanophotonic devices showcases the ability to enhance electric field intensity and adaptability through dielectric layer modulation [42]. In the realm of terahertz (THz) optical systems, chalcogenide metamaterials demonstrate high modulation efficiency and rapid switching capabilities, suggesting a leap forward in THz modulator applications [43]. This is complemented by advancements in dynamic metasurfaces and metadevices, where graphene-enhanced structures illustrate a transition towards electrically controlled electromagnetic wave manipulation, heralding a new era of dynamic functionality [44]. A graphene-based far-infrared absorber introduces adjustable absorption peaks and high sensitivity, an advancing multi-band, a tunable narrowband perfect absorption technology for sensor, photothermal detection, and thermal radiation applications [45]. Finally, the design of a tunable absorption film using AlCuFe quasicrystals demonstrates polarization incoherence and high refractive index sensitivity, promising versatility across diverse applications [46]. Collectively, these studies highlight the dynamic evolution of metamaterials, showcasing their capacity to revolutionize optical and electromagnetic technologies through innovative design and application.

In this context, in this paper, we present simulations and experiments on a novel Nano-Hemisphere on HMM (NHoHMM) structure, combining Ag nano-hemispheres and HMM. We demonstrate the applicability of this structure as an optical absorber. Additionally, we propose a Nano-Hemisphere on Nano-Cermet (NHoNC) structure, combining Ag nano-hemispheres and nanocermet structures, and evaluate its optical properties through simulations and experiments. This study introduces groundbreaking structures that harness the unique properties aiming to extend the functionality and applicability of

plasmonic metamaterials. By presenting these novel structures, we not only bridge the gap between sophisticated plasmonic phenomena and practical applications but also pave the way for advancements in optical absorbers and beyond. These structures leverage the synergistic effects of surface plasmon resonance and material composition to achieve exceptional optical properties, offering new pathways for enhanced light manipulation at the nanoscale. Our findings underscore the potential of these novel configurations in pushing the boundaries of metamaterial applications, marking a significant contribution to the field by introducing structures that can be feasibly produced while providing high performance and versatile functionality.

## 2. Methods

### 2.1. Electromagnetic Field Analysis Simulation

The electromagnetic field analysis simulations were performed using the Finite Difference Time Domain (FDTD) method with commercial software (Poynting for Optics V3L10, Fujitsu, Japan). The model used for the simulation is shown in Figure 1: the thickness of both Ag and $SiO_2$ is 10 nm; the diameter of the Ag nano-hemisphere is 50 nm; and the number of $SiO_2$ and Ag layers are 5 and 4, respectively. Periodic boundary conditions were set in the x and y directions, and absorbing boundary conditions were set in the z direction. The model size for the FDTD calculations using periodic structures was set to twice the diameter. Pulsed light consisting of a differential Gaussian function with a pulse width of 0.5 fs was irradiated from the excitation area as the excitation light. The peak position on the excitation pulse spectrum was approximately 600 THz (corresponding to a wavelength of 500 nm). The reflection and transmission spectra were detected in the reflection detection area and transmission detection area, respectively. The excitation light was X-polarized. The silver nano-hemispherical structure is not anisotropic in the XY direction, so there is no polarization dependence when the light is incident perpendicularly. The dielectric function of Ag was approximated by Drude's equation on the basis of the values reported by Johnson and Christy [22]. The refractive index of $SiO_2$ was set to 1.5 to avoid dispersion. We used a nonuniform mesh with a grid size ranging from 1–5 nm in the x-y direction and 0.5–1 nm in the z direction.

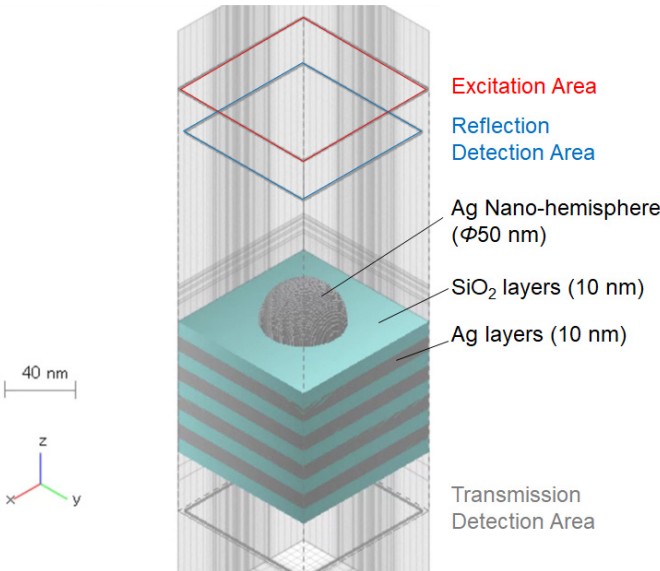

**Figure 1.** Model of NHoHMM structures for the FDTD simulation.

### 2.2. Sample Preparations and Measuremets

Samples of NHoHMM structures were prepared as follows: Using a multi-target sputtering system (FRS-2CP-260T, FKD Factory, Tokyo, Japan), alternating deposition of $SiO_2$ and Ag with the same thickness (10 nm) was performed on a BK7 glass substrate,

creating an Ag/SiO$_2$ multilayer structure. To form the film evenly, the glass substrate was rotated at 10 rpm, ensuring that SiO$_2$ became the topmost layer of the multilayer structure. Subsequently, a 5 nm thick layer of Ag was deposited on top of the multilayer structure, and Ag nano-hemispheres were fabricated through a 300 °C, 10 min heat treatment in a nitrogen atmosphere by a Muffle furnace (F0110, Yamato Scientific Co., Ltd., Tokyo, Japan). Figure 2 illustrates the schematic representation of this process. The size of the Ag nano-hemispheres produced by this method can be adjusted by the initial film thickness and heating temperature. In the present study, the nano-hemisphere structure was fabricated under the same conditions as in the literature [28] to produce a nano-hemisphere structure with a diameter of about 100 nm to have localized surface plasmon resonance in the green well and wavelength region.

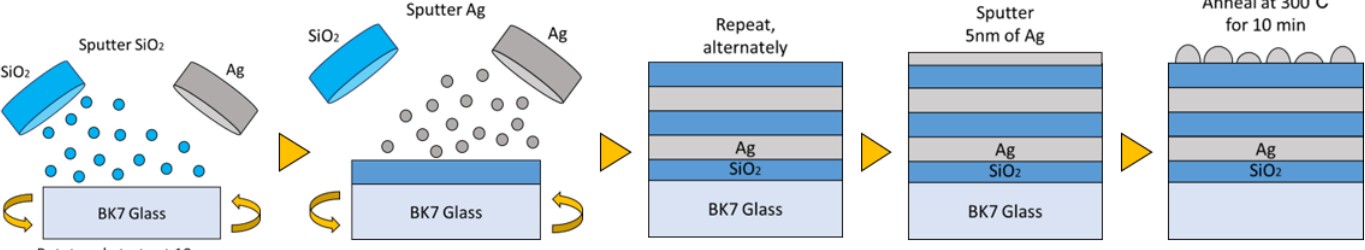

**Figure 2.** Preparation of NHoHMM structures.

The transmission and reflection spectra were measured using a UV–visible spectrometer with a reflectance measurement attachment (5° incident angle, UV-2600, Shimadzu, Kyoto, Japan). The surface structure was evaluated by the scanning electron microscope (SEM) (FlexSEM 1000 II, Hitachi High-Tech, Tokyo, Japan).

## 3. Results and Discussion

### 3.1. Optical Properties of NHoHMM Structures

Figure 3a shows the spatial distribution of the localized electric field of the NHoHMM structure after 3.13, 3.95, and 5.40 fs of excitation obtained through the simulation. The near-field light generated by the Ag nano-hemisphere progresses into the interior of the multilayer structure through the Ag/SiO$_2$ interface. The speed of light propagation due to the localized electric field is reduced to 1/40th of the speed of light propagating in normal vacuum space, indicating that slow light propagation is caused by a penetration of the localized electric field of the LSPR. This is a very characteristic property of the NHoHMM structure, which significantly increases the effective refractive index and is expected to have a strong optical confinement effect. The dynamics of the slow propagation of the localized electric field should be demonstrated in the future by ultrafast optical measurements using long- and short-pulse laser spectroscopy. In Figure 3b, the reflection and transmission spectra obtained through the simulation for the NHoHMM structure are presented. Due to the influence of the Ag layer and the Ag nano-hemisphere, the transmission is nearly zero. In the reflection spectrum, absorption peaks are observed at 380 nm, 400 nm, 450 nm, 600 nm, 660 nm, and 720 nm. A diagram of the electric field distribution at each peak wavelength is shown in Figure 3c. It can be seen that complex modes are formed at each wavelength, but it is difficult to elucidate the details of each. For both modes, it can be seen that the LSPR mode localized in the silver nanoparticles and the hyperbolic metamaterial metal/dielectric multilayer interact with each other.

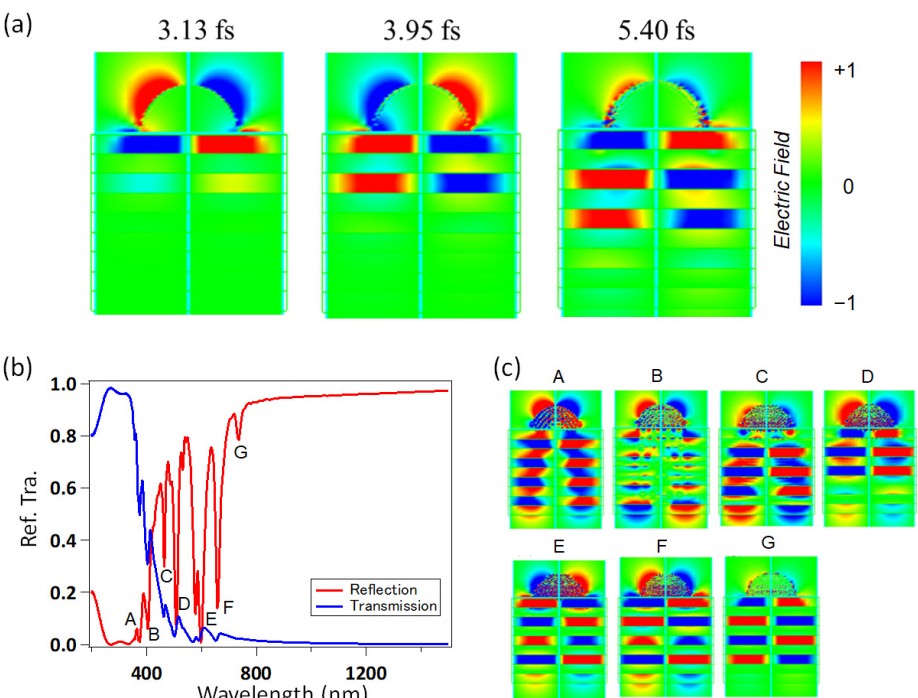

**Figure 3.** (**a**) The spatial distribution of the localized electric field of the NHoHMM structure obtained from the simulation and (**b**) the reflection and transmission spectra. (**c**) Electrical field distributions for each mode of the peaks marked A to G appearing in (**b**).

Figure 4a–d shows the reflection and transmission spectra when varying the number of layers in the multilayer structure. The simulation utilized the structure shown in the inset of the graphs. Figure 4d represents the same structure as used in Figure 3c. From the computational results, it is evident that increasing the number of layers of both Ag and SiO$_2$ leads to an increase in the number of absorption peaks in the reflection spectrum. These results suggest that the absorption peaks are due to surface plasmon resonance occurring at the interface between Ag and SiO$_2$. The simulation results indicate that the NHoHMM structure allows for an easy adjustment of its optical properties by varying the number of layers in the multilayer structure.

An SEM image of the fabricated NHoHMM structure is shown in Figure 5a. Although the arrangement is random and there are variations in size, it can be seen that a nanostructure of an approximately 100 nm system is formed. Attempts were made to observe the metal/dielectric multilayer structure from the cross-sectional SEM, but the multilayer structure could not be observed due to resolution problems. Figure 5b shows the reflection spectra of the NHoHMM structure obtained from the FDTD simulation and the fabricated sample. The structural parameters of the calculation model and the fabricated sample are the same as those in Figure 4a. However, the Ag nano-hemispheres in the fabricated sample have a random structure. The reflection spectrum of the fabricated NHoHMM structure shows only one broad absorption peak instead of the multiple absorption peaks observed in the simulation results. We have already reported that the transmission and reflection spectra of random Ag nano-hemispherical structures fabricated by this method reproduce the peak positions but have a much broader spectral width than the results of FDTD calculations assuming the same size [25–28]. In this case, each resonance peak predicted by the FDTD calculation was adjacent to each other, and the line width of each of them was wider than the peak spacing, which is thought to have resulted in a single broad peak. So, the broad peak obtained experimentally in Figure 5b is roughly consistent with the envelope of the sharp multi-peaks obtained in the calculation.

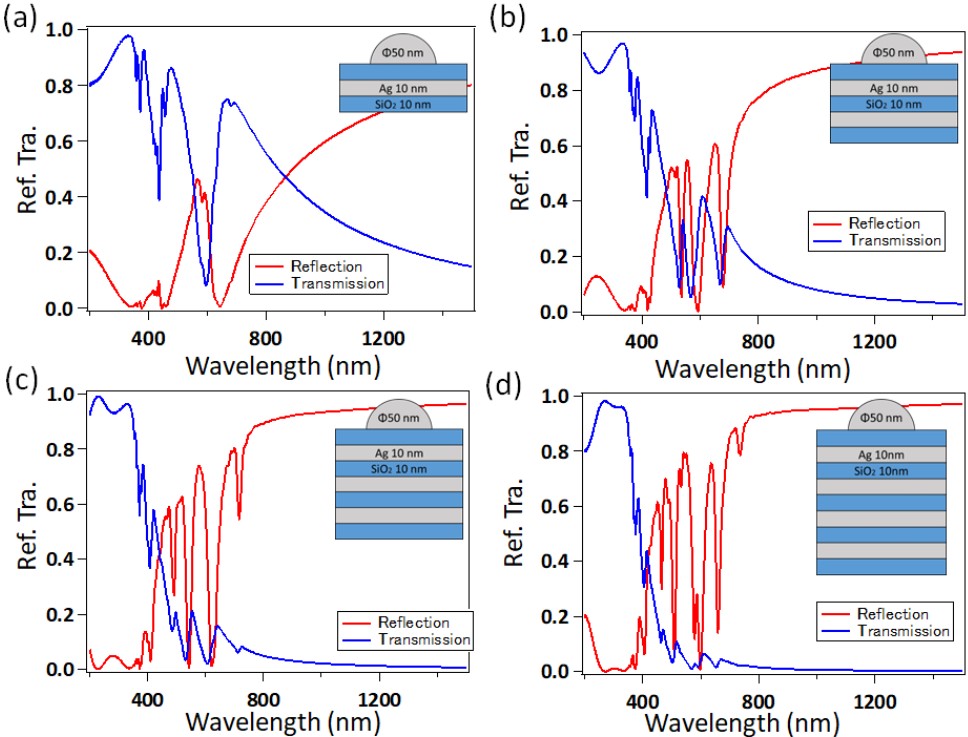

**Figure 4.** Reflection and transmission spectra of the NHoHMM structure when the number of layers in the multilayer structure is varied. The reflection and transmission spectra are shown for Ag/SiO$_2$ with (**a**) 1 layer/2 layers, (**b**) 2 layers/3 layers, (**c**) 3 layers/4 layers, and (**d**) 4 layers/5 layers, respectively.

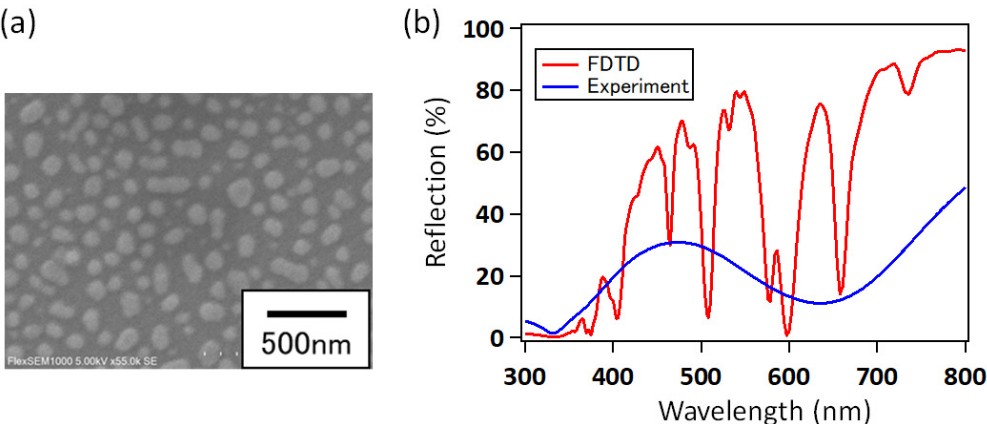

**Figure 5.** (**a**) Scanning Electron Microscope (SEM) image of NHoHMM structure. (**b**) Reflection spectra of NHoHMM structure obtained by experiment and simulation.

Figure 6a–d shows the reflectance spectra of the NHoHMM structure obtained by experiment and simulation when the number of layers in the multilayer structure is varied. The structure shown in the inset of the graph was used in the experiments and simulations. The simulation results show a large change with an increase in the number of layers, while the experimental reflection spectra appear to be almost unchanged. This is thought to be due to the broadening of the absorption peak, which makes the spectral change almost invisible.

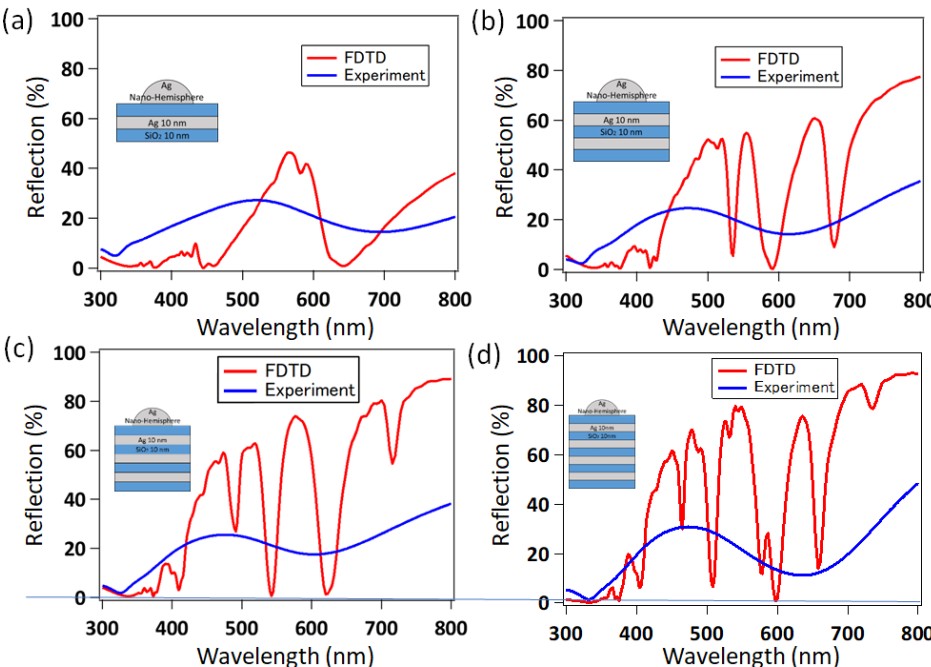

**Figure 6.** Reflection spectra obtained from the fabricated samples and simulation when the number of layers of Ag/SiO$_2$ multilayer structure is varied from (**a**) 1 layers/2 layers, (**b**) 2 layers/3 layers, (**c**) 3 layers/4 layers, and (**d**) 4 layers/5 layers.

To clarify the changes in optical properties, absorption spectra were used in lieu of reflection spectra. The reflection, transmission, and absorption spectra for the number of layers in each multilayer structure are shown in Figure 7a–c. The absorption coefficient *A* was calculated using the following equation based on the reflectance *R* and transmittance *T*:

$$A = 100 - (R + T) \tag{1}$$

As seen in Figure 7, the reflection spectrum remains almost unchanged, but the transmittance decreases significantly as the number of layers increases. The absorption rate also increased significantly with an increase in the number of layers. This is due to an increase in the surface plasmon resonance mode, which is thought to be the same phenomenon as the increase in the absorption peak observed in the simulation results. The fabricated sample did not have multiple sharp absorption peaks as in the simulation results, but rather showed broad absorption spectra over the entire visible region. However, the shape and size of the fabricated nano-hemispherical structures shown in the SEM image in Figure 5a are uniform within a sample area of a few mm and can be fabricated with high reproducibility [25–28]. The reflection and transmission spectra of the structure shown in Figure 5b, which are placed on top of the multilayer, can also be fabricated with good reproducibility. These results suggest that the NHoHMM structure with random Ag nano-hemispheres can be applied as an optical absorber that can be tuned by the parameters of the multilayer structure. For the NHoHMM structure in the present study, no multi-peaks were observed in the experiments, as was the case in the calculations. This may be due to the broadening of the peaks due to the non-uniformity of the hemisphere size, as mentioned earlier, or to the fact that the multilayer structures fabricated by multi-sputtering were intermingled by the heat treatment. A mixed metal/dielectric structure is called a nanocermet structure [36,37], in which metal particles of several nanometres to several tens of nanometres are embedded in dielectric materials. Although the multilayer structure could not be observed in the cross-sectional observation by SEM this time due to resolution problems, the possibility of this structure will be investigated from the optical properties.

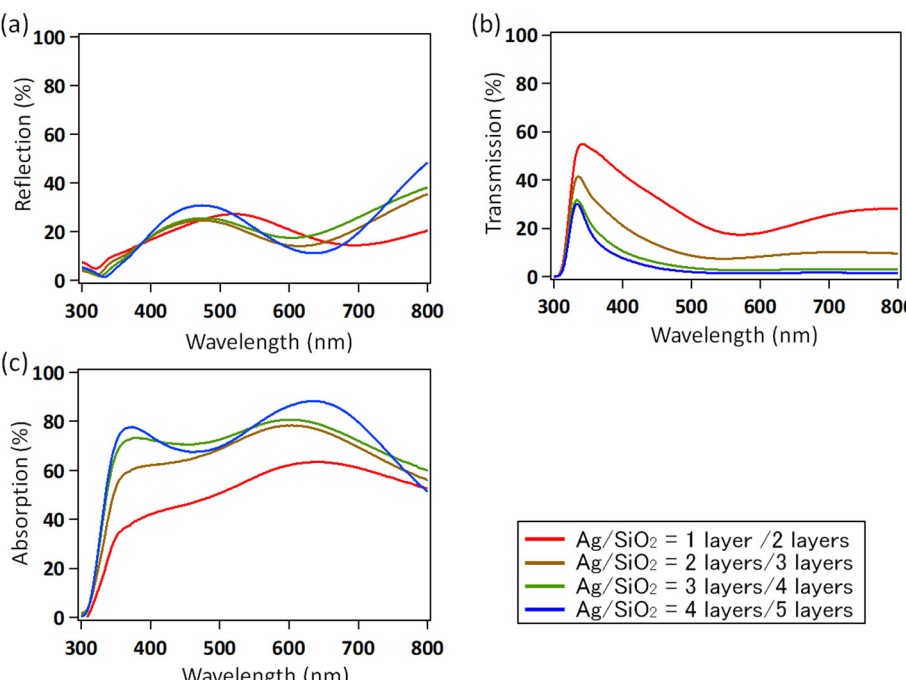

**Figure 7.** (**a**) reflection, (**b**) transmission, and (**c**) absorption of the fabricated NHoHMM structure when the number of layers is varied.

### 3.2. Optical Properties of NHoNC Structure

The optical properties of nanocermet and NHoNC structures were calculated using the transfer matrix method. The nanocermet structure is a complex composite of metal and dielectric, which is difficult to calculate using conventional methods. To compare simulation results, two Effective Medium Theories (EMT) were used: Maxwell Garnett (MG) [47] and Felderhof (FEL) [48]. Both describe the optical properties provided by the presence of metal particles in a dielectric thin film and their LSPR. When the dielectric constant of the matrix is $\varepsilon_d$ and the metallic dielectric constant of the inclusions is $\varepsilon_m$, the effective dielectric constants $\varepsilon_{MG}$ and $\varepsilon_{FEL}$ can be calculated by using MG and FEL, respectively, with the following equations:

$$\varepsilon_{MG} = \frac{\varepsilon_m + 2\varepsilon_d + 2f(\varepsilon_m - \varepsilon_d)}{\varepsilon_m + 2\varepsilon_d - f(\varepsilon_m - \varepsilon_d)} \varepsilon_d \tag{2}$$

$$\varepsilon_{Fel} = \frac{2\varepsilon_d G_s + 3\varepsilon_d}{3 - G_s} \tag{3}$$

$$\left( G_s = \cfrac{3f}{z - \cfrac{a_1}{z + b_2 - \cfrac{a_2}{z + b_3 - \ddots}}} \right) \tag{4}$$

$$\left( z = \frac{\varepsilon_m + 2\varepsilon_d}{\varepsilon_m - \varepsilon_d} \right) \tag{5}$$

The parameters $a_n$ and $b_n$ are given in reference [49] as a function of the filling factor of the inclusions. The dielectric functions of Ag and SiO$_2$ used in calculating the effective dielectric constant are shown below [49,50].

$$\varepsilon_{Ag} = 1 - \frac{\varepsilon_0 \omega_0^2}{\left(\frac{2\pi c}{\lambda}\right)^2 + i\gamma_0 - \omega_0^2} - \frac{\omega_p^2}{\left(\frac{2\pi c}{\lambda}\right)^2 + i\gamma_p\left(\frac{2\pi c}{\lambda}\right)} \tag{6}$$

$$\gamma_p = \gamma_{bulk} + A\frac{v_F}{R}$$

$$\varepsilon_{SiO_2} = 1 + \frac{0.6961663\lambda^2}{\lambda^2 - (0.0684043)^2} + \frac{0.4079426\lambda^2}{\lambda^2 - (0.1162414)^2} + \frac{0.8974794\lambda^2}{\lambda^2 - (9.896161)^2} \tag{7}$$

The dielectric functions of Ag are obtained from the Lorenz–Drude model fitted using the experimental data in Reference [51], with the parameters $\varepsilon_0 = 2.2574$ [rad/s], $\omega_0 = 7.991 \times 10^{16}$ [rad/s], $\gamma_0 = 1.86 \times 10^{14}$ [rad/s], and $\omega_p = 1.366 \times 10^{16}$ [rad/s], respectively. In addition, $\gamma_p$ represents an attenuation constant that takes into account the size effect of silver particles, $\gamma_{bulk} = 4.94 \times 10^{13}$ [rad/s], $v_F = 1.39 \times 10^6$ [rad/s], and $A = 0.5$, and $R$ is the particle radius.

The Ag nano-hemispherical structure was also calculated by assuming it to be a uniform medium using the Effective Medium Approximation (EMA). Thereby, the dielectric function $\varepsilon_{NHS}$ of the Ag nano-hemisphere is expressed by the following equation:

$$\varepsilon_{NHS}(\omega) = \varepsilon_\infty - \frac{\omega_p^2}{\omega^2 - \omega_0^2 - \gamma\omega} \tag{8}$$

The parameters are $\varepsilon_\infty = 3$, $\gamma = 4.5 \times 10^{14}$ [rad/s], $\omega_0 = 3.7 \times 10^{15}$ [rad/s], $\omega_p = 3.5 \times 10^{15}$ [rad/s], respectively. These values were obtained by fitting the results of the FDTD simulation.

The calculation model of the NHoNC structure used in the transfer matrix method consists of four layers of air/Ag nano-hemispheres/nanocermet structure/air, with a homogeneous medium thickness of 30 nm assuming an Ag nano-hemisphere and a homogeneous medium thickness of 50 nm for the nanocermet structure.

Figure 8a,b shows the reflection and transmission spectra of the NHoNC structure obtained by the MG and transfer matrix method. In the calculations, the Ag filling factor in the nanocermet structure varied from 5 to 30%. Both reflection and transmission spectra show peaks at wavelengths of around 400 nm and 500 nm, respectively. This is thought to be due to the surface plasmon resonance generated by the Ag nanoclusters in the nanocermet structure and the Ag nano-hemispheres. As the Ag filling factor increases, these peaks become broader, and a new absorption peak around 520 nm is observed in the reflection spectrum. This absorption peak is thought to be due to the mirror image effect of the nanocermet structure and Ag nano-hemispheres, and the peaks at 400 nm and 500 nm are broadened by the Ag nanoclusters bonding. Figure 8c,d shows the reflection and transmission spectra of the NHoNC structure obtained by the FEL and transfer matrix methods. The trends are generally in agreement with the results using the MG, and both calculation methods are considered to be useful for NHoNC structures.

The Ag/SiO$_2$ nanocermet structure was fabricated by simultaneous deposition of SiO$_2$ and Ag on BK7 glass using a multi-sputtering system. Figure 9 shows this process. In this study, a sample with 20% Ag filling was fabricated.

Figure 10a,b shows the reflection and transmission spectra of the fabricated samples and the NHoNC structure obtained by the transfer matrix method. The Ag filling factor of the nanocermet structure was set to 20% in both cases. Both reflection and transmission spectra of the fabricated NHoNC structures are very broad, and the peak wavelengths are located at longer wavelengths than those predicted by the calculation. This is thought to be due to the randomness of the Ag nano-hemispheres. The peak around 400~500 nm, which was observed in the calculation, could hardly be seen in the experimental results. This is because the Ag nanoclusters in the nanocermet structure tended to aggregate during the fabrication process, which prevented plasmon resonance from occurring in the nanoclusters. In this experiment, a sample with a relatively high Ag content of 20% was used, but sufficient light absorption could be obtained by using a structure with a lower Ag filling factor.

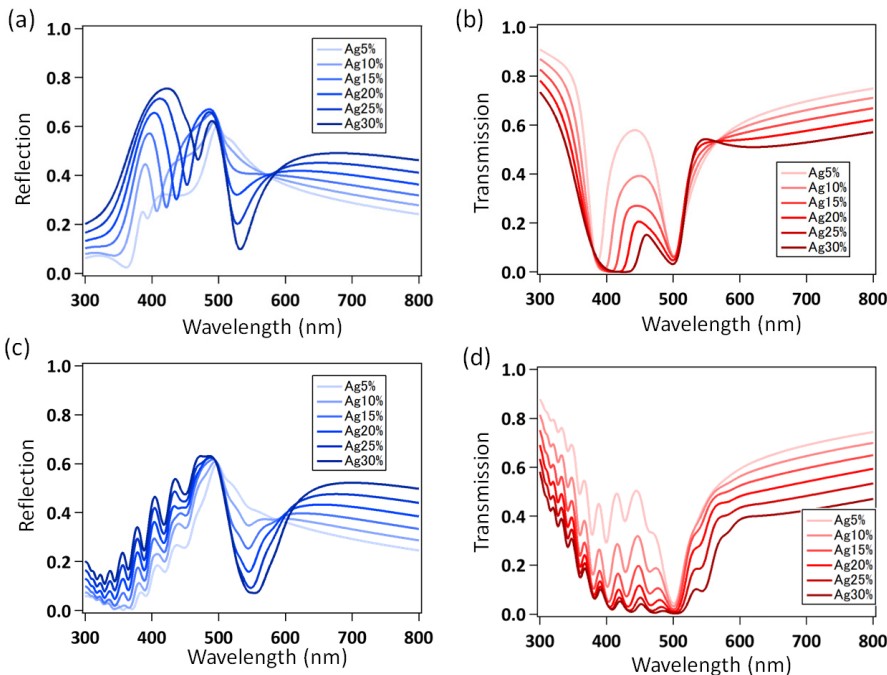

**Figure 8.** (**a**) Reflection and (**b**) transmission spectra of the NHoNC structure obtained by the MG and the transfer matrix method. The legend indicates the Ag filling factor in the nanocermet structure. (**c**) Reflection and (**d**) transmission spectra of the NHoNC structure obtained by the FEL and the transfer matrix method.

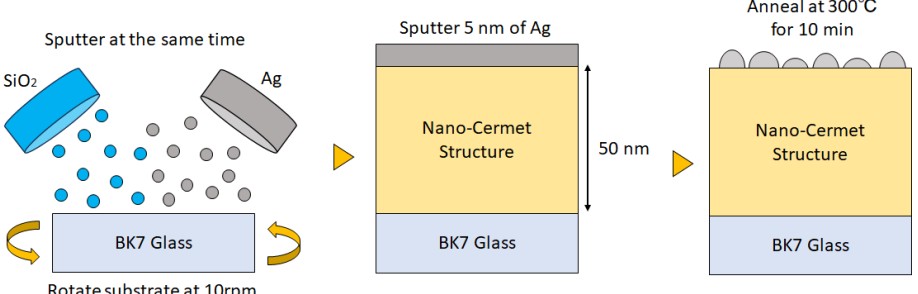

**Figure 9.** Preparation of NHoNC structures.

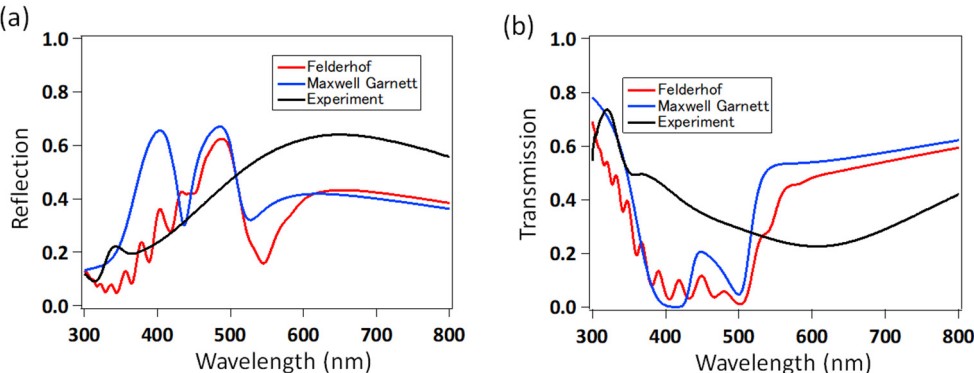

**Figure 10.** (**a**) Reflection and (**b**) transmission of the NHoNC structure obtained from the fabricated samples and the transfer matrix method.

Comparing the transmission and reflection spectra of the NHoNC structure shown in Figure 10a,b with the transmission and reflection spectra of the NHoHMM structure shown in Figure 6a,b, it is observed that the reflection spectra, in particular, differ significantly.

Therefore, the NHoHMM structure should differ from the cermet structure. This suggests that the reason why the NHoHMM structure does not exhibit multiple peaks as predicted by calculations but rather broad peaks, is not because the dielectric/metal multilayers have broken down and mixed together to form a cermet-like structure. Although not yet confirmed directly by high-resolution structural observations, we believe that the hyperbolic metamaterials with metal/dielectric multilayers were formed, and the peaks became broader and overlapped each other due to the inhomogeneity of layer thicknesses of layers and the sizes of the Ag nanoparticles.

## 4. Conclusions

We report a novel metamaterial, the NHoHMM structure, which combines Ag nano-hemispheres and metal/dielectric composite structures. This structure can be fabricated using only two steps, annealing and sputtering, and does not require a top-down nanofabrication process. The NHoHMM structure consists of Ag nano-hemispheres fabricated on an Ag/SiO$_2$ multilayer structure. The FDTD simulation results showed the slow propagation of the localized electric field and multiple absorption peaks, which are caused by surface plasmon resonance at the Ag/SiO$_2$ interface, and the optical properties can be altered by changing the number of layers in the multilayer structure. In the experiment, no multiple absorption peaks were observed, and broad absorption peaks were obtained over the entire visible region. The absorption coefficient increased as the number of layers was increased. Therefore, the NHoHMM structure has the potential to achieve highly efficient optical absorption.

For comparison, the NHoNC structure consists of Ag nano-hemispheres fabricated on an Ag/SiO$_2$ nanocermet structure, and the reflection and transmission spectra were calculated using the effective medium theory and the transfer matrix method. As a result, absorption peaks due to Ag nanoclusters and Ag nano-hemispheres were observed at wavelengths of around 400~500 nm. The interaction between the Ag nano-hemispheres and the nanocermet structure increased with an increasing Ag filling factor, and an absorption peak due to the interaction was observed at a wavelength of 520 nm. In the experiment, the calculated peak was not observed, and only one broad peak appeared at the longer wavelength side. This is due to the random Ag nano-hemisphere structure, and the nanoclusters were not well-formed in the nanocermet structure fabricated in this study due to the high Ag filling factor. It is expected that the fabrication of NHoNC structures with a lower Ag filling factor will produce the expected optical absorption. Although the formation of the multilayer structure of the NHoHMM must be confirmed by a high-resolution cross-sectional TEM, it is clear that the optical properties of the NHoHMM structure differ from those of the NHoNC structure.

To confirm the multi-peaks obtained by simulation, future work will need to involve fabricating NHoHMM structures of uniform size and periodic arrangement using top-down processes such as electron beam lithography and FIB. The slow propagation of the localized electric field observed in the simulations should also be demonstrated using ultrashort-pulse laser spectroscopy. The properties of these NHoHMM structures indicate an extremely strong optical confinement effect and are expected to be advantageous for use as a substrate material in highly sensitive sensing and energy devices through efficient optical harvesting. They are also anticipated to benefit highly integrated optical waveguides, optical circuits, and optical computing based on such devices. To extend these applications, it will be crucial to optimize the nanostructure in the future to maximize the absorption and optical confinement of this structure.

**Author Contributions:** Conceptualization, methodology, and software, R.N. and K.O.; validation, T.M. and K.W.; investigation, resources, and data curation, R.N.; writing—original draft presentation, R.N.; writing—review and editing, K.O.; supervision, project administration, and funding acquisition, K.O. All authors have read and agreed to the published version of the manuscript.

**Funding:** This work was supported by the JSPS Grants-in-Aid for Specially Promoted Research (No. JP20H05622) and Scientific Research (S) (No. JP19H05627).

**Institutional Review Board Statement:** Not applicable.

**Informed Consent Statement:** Not applicable.

**Data Availability Statement:** The data presented in this study are available on request from the corresponding author.

**Conflicts of Interest:** The funders had no role in the design of the study, in the collection, analyses, or interpretation of data, in the writing of the manuscript, or in the decision to publish the results.

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
