# Peer review of "Novel Plasmonic Metamaterials Based on Metal Nano-Hemispheres and Metal-Dielectric Composites"

_photonics, doi:10.3390/photonics11040356_

Round 1

Reviewer 1 Report

Comments and Suggestions for Authors

The authors both numerically and experimentally proposed novel plasmonic metamaterials based on metal nano-hemi-spheres and metal-dielectric composites. Taking into account the obtained results, I would like the authors to comment in detail on the following issues:

1.     Some functionalities of hyperbolic metamaterials appear in the introduction. However, it is worth taking a broader look at this issue and referring to several recently introduced concepts, e.g.: Sci Rep 11, 74 (2021); Sci Rep 12, 16961 (2022), Applied Sciences 8.8 (2018): 1222, etc.

2.     I wonder what is the point of creating a numerical model and simulating the properties of the metamaterial if this model does not correspond to or reproduce the experimental sample. As a result, we observe overlapping reflectance results from simulations and experimental tests, the correlation of which is practically non-existent. In my opinion this is pointless.

3.     The authors' claim about the slow light effect requires additional proof. How many picoseconds is it actually? So far - based on the presented results - we only see the plasmon-induced transparency (PIT) effect. To fully explain these phenomena, it is necessary to make spatial distributions of the electric field intensity within the presented structure for the wavelength where the peak is observed. Then the transmission phase shift and group index should be calculated. Without such results, talking about this effect is a bit exaggerated.

My conclusion is that the article may be interesting, but many issues require clarification and expansion. I suggest the authors show the optimized device, because proving that increasing the stack size causes the appearance of new transmission/reflection peaks does not contribute much to the current knowledge about hyperbolic metamaterials. I would focus the manuscript and present the properties of the hyperbolic metamaterial in the form of a stack with optimized parameters, i.e. maximum absorption, and in the case of the second configuration, I would expand the results and reliable explanations of the PIT and the slow light effect.

Comments on the Quality of English Language

Minor editing of English language required

Reviewer 2 Report

Comments and Suggestions for Authors

The manuscript ID photonics-2874787 mainly presents a study about two different plasmonic metamaterials based on metal nano-hemi-spheres and metal-dielectric composites. The silver-based systems correspond to the assistance of a hyperbolic metamaterial or a nano-cermet structure. Numerical and experimental results are presented. Please see below some points to the authors:

  1. No match is present in the numerical and experimental data related to reflection and transmission plotted in figures 4, 5 and 11. Something about this issue should be improved.
  2. A numerical fitting in the absorbance spectra of the samples should be included.
  3. A high resolution micrograph of the plasmonic metamaterials would be welcome.
  4. Do the transmittance is sensitive to incident polarization?
  5. Plasmonic properties can be modulated by small variations in the angle of incidence, which can be provided by nonlinear effects induced by femtosecond pulses. The authors are invited to discuss this issue in their experiments and contrast with other plasmonic materials. You can see for instance: https://doi.org/10.1364/JOSAB.32.000805
  6. The discussion section should be importantly improved. The main findings should be confronted with updated publications in the topic in order to highlight the value of the results for the application proposed. You can see for instance: https://doi.org/10.1016/j.optcom.2019.07.019
  7. It is not clear how was selected the size of the nanostructures in order to see that the work is systematic instead of incidental.
  8. Statistics in the experiments and reproducibility in the preparation of the samples ought to be described.
  9. How is carried out the annealing of the samples in order to guarantee an isotropic response of the system derived from the processing route. Evidences of the homogeneity of the samples should be provided.
  10. The citations presented in collective form should be split in order to better present the individual importance of the references selected for this topic.
Comments on the Quality of English Language

A proofreading is suggested

Reviewer 3 Report

Comments and Suggestions for Authors

In this paper, the authors report two types of plasma metamaterials: nanospheres (NHoHMM) on hyperbolic metamaterials and nanospheres (NHoNC) on nano metal ceramics. Simulations reveal distinctive sharp absorption peaks in the visible spectrum, attributed to surface plasmon resonance. In practical experiments, the NHoHMM structure, characterized by random Ag nano-hemispheres, exhibits broad absorption peaks spanning the visible range, rendering it a versatile broadband optical absorber. I believe that publication of the manuscript may be considered only after the following issues have been resolved.

1.    I don't quite understand the experimental results mentioned in the article, did the author conduct relevant experiments? Suggest the author to provide relevant sample testing data, such as SEM.

2.    Section 2.1 of the article is too simplistic. The author needs to provide specific and detailed simulation parameters for relevant readers to repeat. Suggest the author to provide relevant simulation structure diagrams.

3.    What is the physical mechanism behind the appearance of relevant optical properties in this structure? Suggest the author to provide necessary electromagnetic field simulation results.

4.    All numbers and units in the article need to be left blank. Including figures, tables, and the main text.

5.    In the introduction section, regarding plasmonic metamaterials, the author needs to mention some of the latest related work, such as, Optics & Laser Technology 169, 2024, 110186; Opto-Electron Sci 1, 210010 (2022); Opto-Electron Adv 5, 200098 (2022); Diamond and Related Materials, 136, 2023, 109960.

6.    The English expression of the whole article needs to be further improved.

Comments on the Quality of English Language

Minor editing of English language required

Round 2

Reviewer 1 Report

Comments and Suggestions for Authors

The authors revised the manuscript in accordance with my earlier review. Therefore, I believe that the manuscript can be published in its current version.

Reviewer 2 Report

Comments and Suggestions for Authors

The authors have provided a clear reply to the points raised in the review stage. This study about two different plasmonic metamaterials based on metal nano-hemi-spheres and metal-dielectric composites is interesting and it can be useful for future research. Then, I can recommend this reviewed manuscript for publication in present form.

Comments on the Quality of English Language

A proofreading is suggested

Reviewer 3 Report

Comments and Suggestions for Authors

Accept in present form